# Effects of Growth Hormone (GH) Supplementation on Dermatoscopic Evolution of Pigmentary Lesions in Children with Growth Hormone Deficiency (GHD)

**DOI:** 10.3390/jcm11030736

**Published:** 2022-01-29

**Authors:** Fabrizio Panarese, Giulio Gualdi, Marta Di Nicola, Cosimo Giannini, Nella Polidori, Federica Giuliani, Angelika Mohn, Paolo Amerio

**Affiliations:** 1Department of Dermatology, University “G D’Annunzio” University of Chieti-Pescara, 66100 Chieti, Italy; fabrizio.panarese@gmail.com (F.P.); federicagiuliani@hotmail.it (F.G.); 2Department of Experimental and Clinical Sciences, Biostatistic Laboratory, University “G D’Annunzio” University of Chieti-Pescara, 66100 Chieti, Italy; m.dinicola@unich.it; 3Department of Pediatrics, University “G D’Annunzio” University of Chieti-Pescara, 66100 Chieti, Italy; cosimogiannini@hotmail.it (C.G.); nella.polidori@hotmail.com (N.P.); angelica.Mohn@unich.it (A.M.)

**Keywords:** nevi, GH, melanoma

## Abstract

Recent reports have confirmed higher levels of growth hormone (GH) receptor (GHR) transcripts in malignant melanomas (MM), yet the role of GH in the pathogenesis of MM remains controversial. Although melanocytes appear to be hormonally responsive, the effects of GH on MM cells are less clear. A direct correlation between GH administration and the development of melanoma seems possible. Our study aimed to assess whether GH supplementation in children with growth hormone deficiency (GHD) could induce changes in the melanocytic lesions both from a dimensional and dermoscopic point of view. The study population consisted of 14 patients sorted into two groups. The experimental group consisted of seven GHD pediatric patients who underwent dermatological examination with epiluminescence through the use of digital video recording of all melanocytic lesions before and after 12 months of GH supplementation, whilst the control group consisted of seven healthy pediatric patients matched for age, sex and phototype. All patients were evaluated according to auxological and dermatological features. A total of 225 melanocytic lesions were examined in the experimental group and 236 in the control group. Our study shows a significant increase in the mean size values of the lesions in the study group but not in the control group. Increases in the dermoscopic ABCD Score and in BMI correlated to an increase in the size of the melanocytic lesions and the dermoscopic parameters. The increase in SDS Height correlated with ABCD Score changes and with dermoscopic score structures. No differences were found compared to the control group. Dimensional/structural modifications in melanocytic lesions of patients treated with GH were closely related to weight and statural growth and can be considered a normal physiological process induced by GH supplementation.

## 1. Introduction

Melanoma originates from melanocytes that have switched to cancerous cells as a consequence of aberrant changes at molecular and biochemical levels [1]. Melanoma is characterized by an extensive degree of heterogeneity in terms of clinical, dermatological and histopathological presentation [2]; genomic and post-genomic profiles [3,4,5,6]; and risk factors (skin type, exposure to sun radiation, number of nevi, age, gender, immune status, family history or history of previously removed melanomas). This tumor is characterized by a high propensity to metastasize [7,8], making this disorder a significant public health issue. Concerning the incidence of melanoma, gender disparities have been reported, with an increased incidence in men. This difference can be attributed to physiological differences in skin structure, baseline differences in immune systems, the influence of sex hormone levels and estrogen receptor expression [9,10,11,12]. Over the last few decades, the tumor-driving properties of GH and GHR have been established for breast, colon and prostate cancer [13,14,15,16]. The first identification of growth hormone receptor (GHR) RNA in human skin melanocyte cells dates back more than 20 years [17] and has been followed by the discovery of autocrine levels of GH and IGF1, also present in normal skin and basal cell carcinoma [17,18]. Several studies report elevated levels of RNA and GHR proteins in human melanoma biopsies [19,20,21,22,23], while the cell cycle of melanoma cells has been shown to be subject to orchestrated regulation of endogenous GH, prolactin (PRL) and adrenocorticotropic hormone (ACTH) [24]. Furthermore, in primary human melanoma samples, an elevated expression of the GH-releasing hormone receptor (GHRH) (GHRHR) has been demonstrated [25], while GHRH analogues have been shown to suppress the growth of malignant melanomas in vivo [26].

It is well established that pigmentation is regulated by a variety of signaling pathways, including peptide hormones [27]. Consequently, a direct correlation between exogenous GH administration and the development of melanoma seems possible. This correlation may also extend to benign melanocytic neoplasms. A previous study supports this idea, demonstrating that the administration of exogenous GH accelerates the rate of nevus growth [28]. There is also evidence that nevi of individuals treated with GH exhibit abnormal melanosome architecture, increased Ki-67 staining and increased anisokaryosis when compared with matched controls [28]. However, it should be noted that other studies have failed to confirm this correlation [20,29,30,31,32]. To date, there are no clinical studies that have documented changes in the growth of single melanocytic lesions in patients treated with GH by using digital dermatoscopy. Through this non-invasive method, which have been previously utilized to examine melanocytic lesions subjected to photoexposure, we were able to easily monitor possible activation signals and examine potential associations between GH supplementation and melanocytic lesions [32].

## 2. Materials and Methods

Seven children with a new diagnosis of GHD were recruited (4 males and 3 females), all of whom were attending the outpatient clinic for growth disorders at the Pediatric department of the Santissima Annunziata in Chieti. A comparable control group of 7 healthy pediatric patients, matched for age, sex, skin phenotype and BMI, was also recruited. Before commencing GH supplementation therapy at a dosage of 0.7 mg/kg/day, the patients underwent a dermatological examination with epiluminescence. A digital video recording of all the melanocytic lesions was carried out. In order to reduce the influence of sun exposure, melanocytic lesions located on chronically photoexposed areas (face, scalp and hands) were excluded. For each participant, a consent form and a questionnaire were completed, with the latter collecting data about age, sex, skin phototype, history of familial melanoma, sun sensitivity, sun exposure and previous sunburn history. After 12 months of therapy (T1), all patients were re-evaluated from both an auxological and a dermatological point of view. For each nevus, dermoscopic images were stored at T0 and T1 using the Fotofinder dermatoscope (Fotofinder Systems GmbH, Bad Birnbach, Germany). Dimensions were measured in millimeters at the two major axes (AsseMAX and AsseMIN). A dermoscopic score was assigned to each image, according to the Stoltz dermoscopic ABCD criteria, based on symmetry, edges, color and dermoscopic structures. The documented lesions were blindly scored by two different dermatologists who are experts in dermoscopy (GG and FP). We calculated the average of the values measured by the two dermoscopists for each of the variables and for the final total score (ABCD SCORE). Auxological variables recorded were as follows: Body Mass Index (body weight/height ^2^), used as an indirect measure of whole-body growth; and Standard Deviation Score for height (SDS Height), used as an objective measure of the height reached by the subject and therefore a direct measure of GH efficacy in boosting subject growth. IGF-1 blood levels were recorded before and after 1 year of GH treatment as a measure of treatment efficacy. These parameters were described through the median and the interquartile range (IQR) due to the non-normal distribution of the data. The nevus-related variables, meanwhile, were instead described using the mean and standard deviation (SD). The comparison between baseline and follow-up values for each of the auxological variables was performed using the non-parametric Wilcoxon test for dependent samples. The comparison between baseline and follow-up values for each of the nevus-related variables was performed using the Student’s t test for 30 paired samples. Correlations between the variation in the SCORE ABCD, the variation in the maximum diameter of the nevi, the variation in the minimum diameter of the nevi and the variation in each of the analyzed auxological variables were analyzed. All analyses were performed using IBM SPSS Statistics 20 software (Armonk, NY, USA).

## 3. Results

The average age of all children enrolled in the study was 11.1 years, with an average of 10.9 for the study group and 11.3 years for the control group. The auxological characteristics of the study patients in terms of median, interquartile range and relative variation between the baseline and follow-up values are reported in Table 1. A total of 456 recorded melanocytic lesions were examined at T0 (222 in the study group, 234 in the control group), increasing to 461 (225 in the study group, 236 in the control group) at T1 because of the appearance of three new pigmented lesions in three of the study patients and two new pigmented lesions in two of the control patients. Regarding the analysis of melanocytic lesions of the study group, the mean and standard deviation of the values relative to the size of the major axis (AsseMAX) measured at the baseline were 2.22 ± 1.07 mm, increasing at the follow-up to 2.35 ± 1.10 mm, with a relative change of 0.07 ± 0.12 mm (*p*-value < 0.001) (Figure 1). Regarding the minor axis (AsseMIN), the baseline values were 1.72 ± 0.85 mm against 1.85 ± 0.88 mm at follow-up, with a relative variation of 0.09 ± 0.13 (*p*-value < 0.001) (Figure 1). Comparison with the control group reveals that the lesions in the control-group subjects were of a greater size at baseline. However, their size increase at follow-up was negligible and not significant (Table 2). Since no significant differences were found in the control group, qualitative analyses were performed only for the study group. The mean and standard deviation of individual dermoscopic parameters (Asymmetry, Edges, Color and Structure) of the Stolz algorithm and the final score values (ABCD Score) relative to the baseline and follow-up, together with absolute variation are shown in Table 3. As far as the ABCD Score values are concerned, the average values at baseline were 1.69 ± 0.84 against the values of 1.96 ± 0.86 in the follow-up control, with an absolute variation of 0.27 ± 0.87 (*p*-value < 0.001) (Figure 2). All the data obtained were correlated with auxological variables. The change in BMI (MIBMI) statistically correlated with each of the following: MAX axis (R = 0.266, *p*-value <0.001) and MIN axis variations (R = 0.142, *p*-value = 0.034); ABCD Score (R = 0.268, *p*-value < 0.001); Edges Score (R = 200, *p*-value = 0.003); Colors Score (R = 0.346, *p*-value < 0.001); and Structures Score (R = 0.157, *p*-value < 0.019). The variation in SDS Height (DSSDS) statistically correlated with the ABCD Score (R = 0.151, *p*-value = 0.025) and with the Structures Score (R = 0.162 *p* = 0.015). Weight variation (ΔW) statically correlated with the Edges Score (R = 0.188, *p*-value = 0.005) and with the Colors Score (R = 0.255, *p*-value < 0.001). These results are summarized in Table 4.

## 4. Discussion

Although there have been studies that highlight a correlation between the GH/IGF axis and mole modification, the literature is limited and difficult to interpret. To date, this is the first study in which the effect of GH supplementation on melanocytic nevi of children with GHD has been dermatoscopically evaluated. Since the study group was composed of a small number of patients, we introduced a control group of healthy patients, matched for age, sex and phenotype, in order to improve the significance of the research.

Our results showed no significant difference in the number of pigmented lesions between the two groups. However, we observed a statistically significant increase in the mean size values of the lesions for the MAX axis (an increase of 0.07 ± 0.12 mm) and for the MIN axis (an increase of 0.09 ± 0.13 mm) for the study group, compared to the control group, even though the latter showed larger moles at baseline (baseline AsseMAX of 2.30 ± 0.47 mm against baseline AsseMIN of 1.79 ± 0.66 mm).

These findings may suggest that GH supplementation works to accelerate the growth of pigmented lesions in a sort of synchronicity between melanocytic maturation and body growth. Such an effect would not be as evident in the control group since the growth rate of these patients during the study period would not be as fast as in the patients with GH supplementation (data not shown). The dermoscopic ABCD Score, which takes into account a variety of melanocytic features in order to assess its potential malignancy, increased by 0.27 ± 0.87 points. Although this result is statistically significant, in our view it does not imply important repercussions for clinical practice, given that the maximum values recorded for the ABCD Score were 4.7 and 4.2 for the T0 and the T1 values, respectively, and the cut-off value for benign melanocytic lesions in dermoscopic ABCD evaluation is 4.75.

Dermatological data were more interesting when evaluated in comparison with the auxological data. In particular, we demonstrated that an increase in BMI, an indirect growth index, corresponded to increases in the MAX Axis, the MIN Axis and the dermoscopic parameters relating to the scores for Edges, Color and Structures and for ABCD overall score. The increase in SDS Height, which indicates an approach to the physiological growth curve, and which could therefore be considered a clinical marker for the efficacy of GH therapy, correlated with the changes in the ABCD Score and with the changes in the score for dermoscopic structures.

The moles’ dimensional changes did not appear to be due to the action of IGF-1, since there was no correlation of the dimensional and dermoscopic parameters of melanocytic lesions with blood levels of IGF-1. Both benign and malignant melanocytes possess receptors for IGF-1 [17,33,34], the main downstream mediator of HGH. IGF-1 has been shown to play a role in the progression of early malignant lesions [35] as well as in promoting their progression and dissemination [36].

In our study we recorded only minimal modifications to the melanocytic lesions, which are not significant from a clinical point of view, but were still present and objectively observable. These can be interpreted as the results of mole activation and melanocytic growth. It is known, for example, that the presence of multiple globules located on the periphery of the lesion is a sign of growth, and that changes in color can be attributed to the migration of the melanocytic elements from deep to superficial planes, or vice versa. In Figure 3 and Figure 4, there are some examples of dermoscopic images at T0 and T1. These dimensional and structural modifications, which are closely related to the weight and statural growth of the patients, can be considered a physiological process induced by treatment with GH supplementation.

## 5. Conclusions

From our data, it appears that GH therapy does not induce drastic changes in moles, and that the effects of such therapy are probably limited to melanocyte activation that can be detected at a dermatoscopic but not at a clinical level (i.e., macroscopically, by eye examination) within one year from the beginning of the treatment. The main limitation of this study is related to the small number of patients and the short 12-month follow-up; a longer follow-up period and a higher number of patients would be necessary to definitively asses GH safety with regard to melanoma formation. 

## Figures and Tables

**Figure 1 jcm-11-00736-f001:**
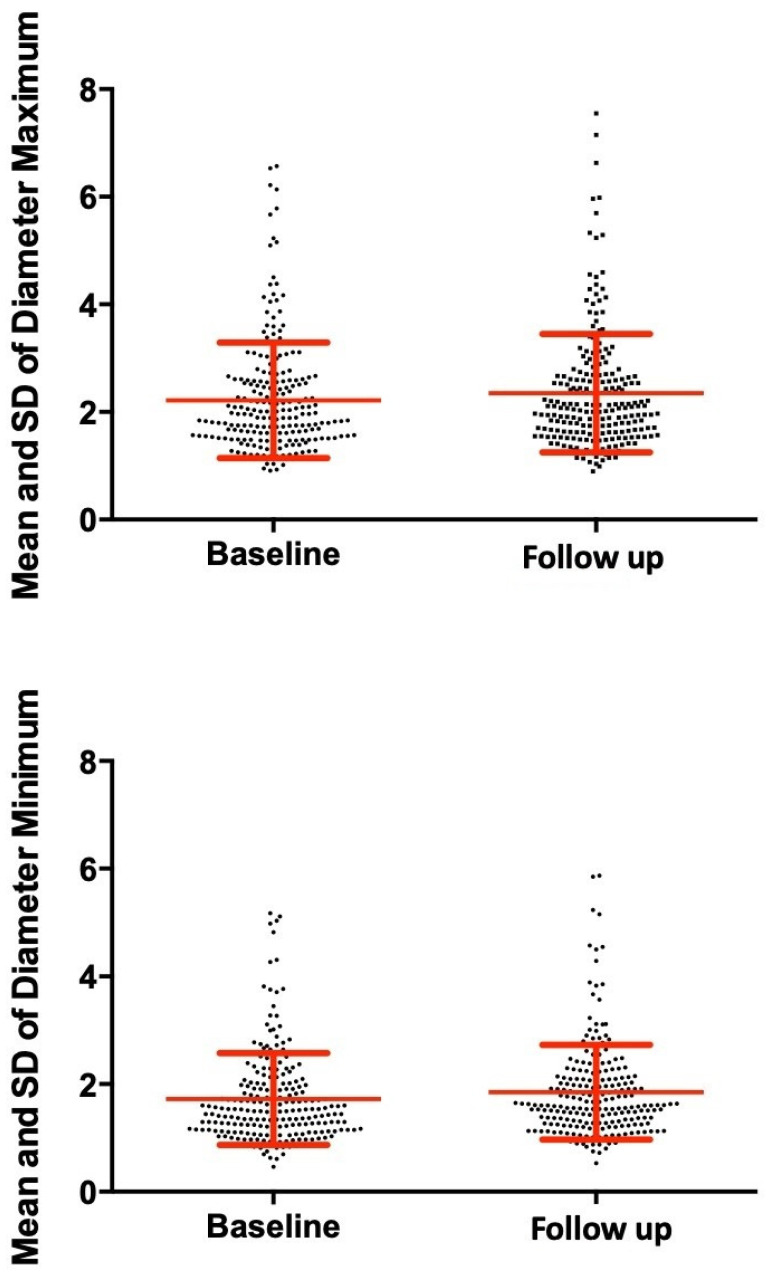
Mean and standard deviation (SD) of diameters (mm) at baseline and follow-up.

**Figure 2 jcm-11-00736-f002:**
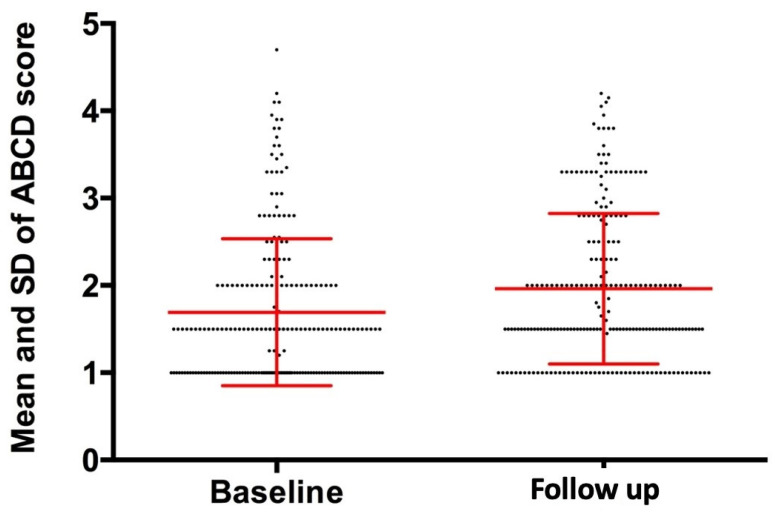
Mean and standard deviation (SD) of ABCD Score at baseline and follow-up.

**Figure 3 jcm-11-00736-f003:**
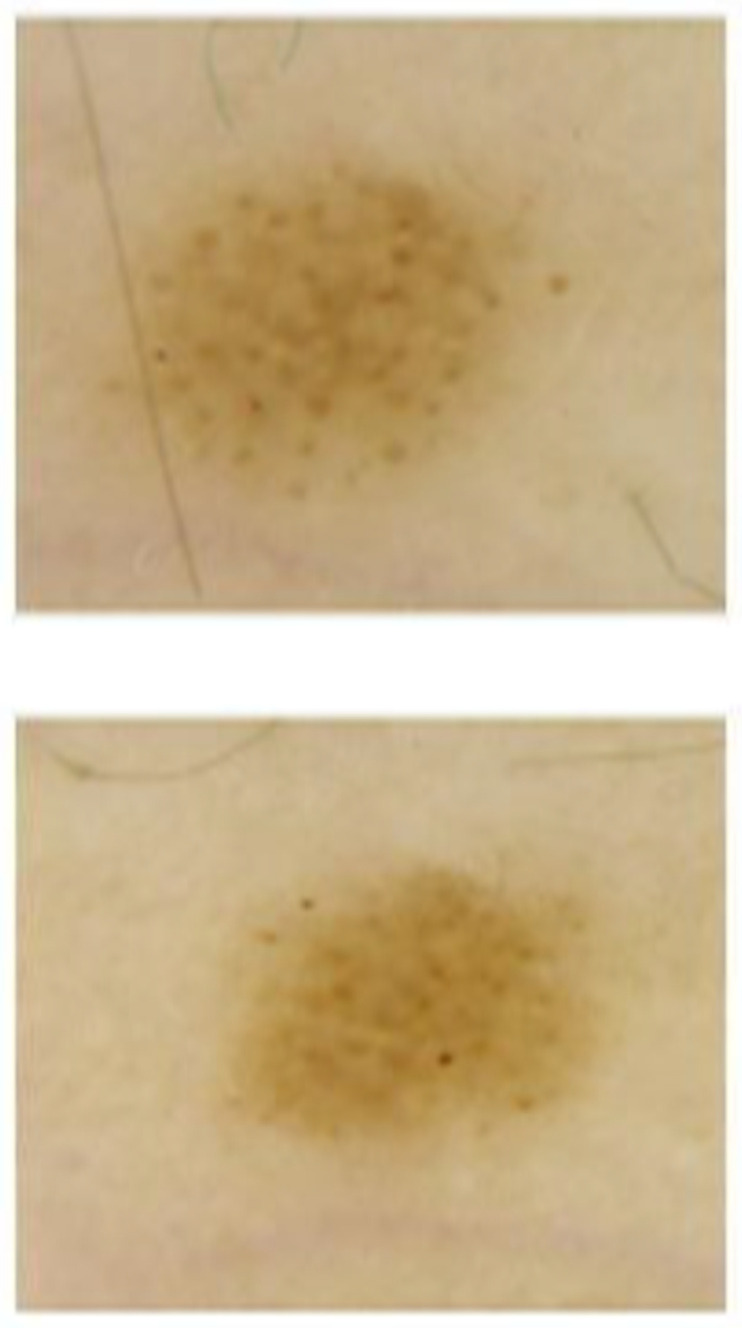
Example of nevi change in experimental group subject at T0 (**lower**) and T1 (**upper**). Expansion of diameters and presence of multiple pigmented globules over the lesion surface are signs of the physiological expansion of the mole.

**Figure 4 jcm-11-00736-f004:**
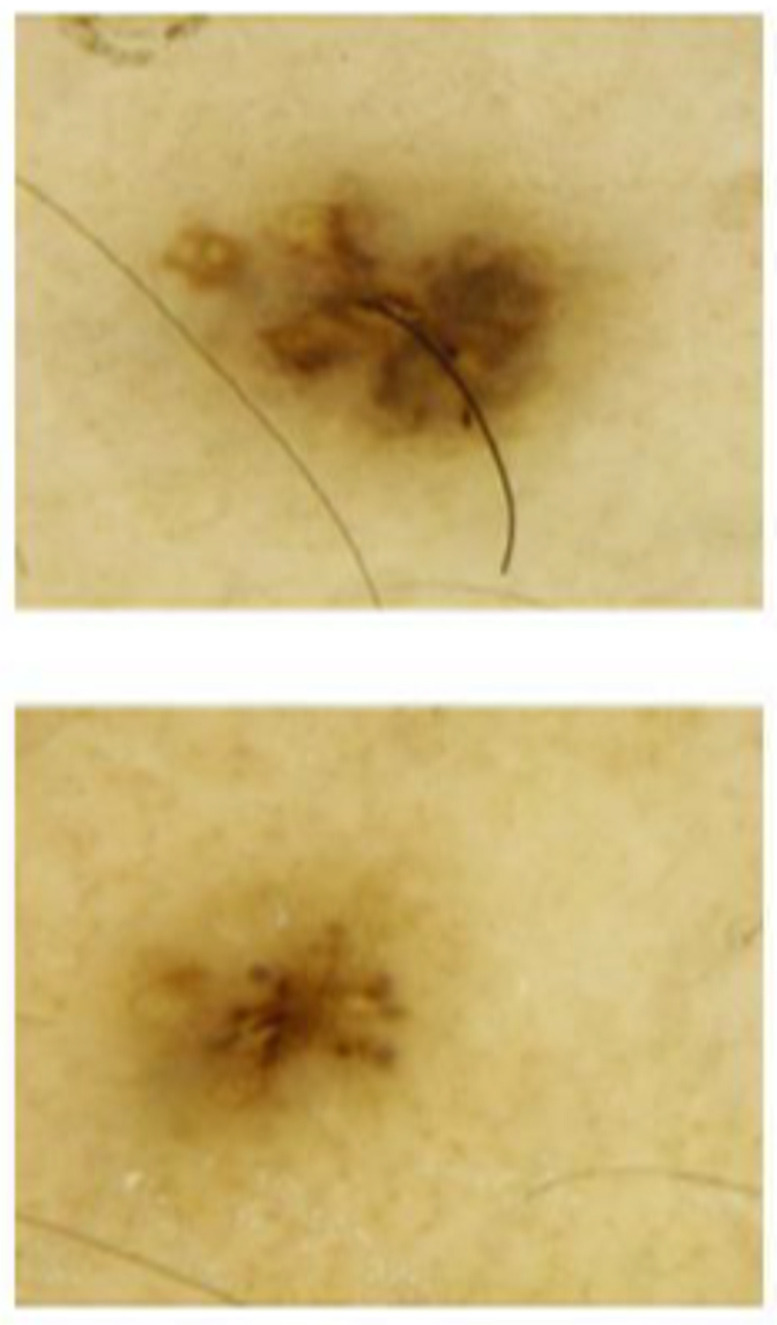
Example of nevi change in experimental group subject at T0 (**lower**) and T1 (**upper**), showing expansion of diameters and presence of multiple pigmented globules located on the periphery of the lesion, indicating the growth of the lesion. The change in color can be attributed to the migration of the melanocytic elements from deep to superficial planes, or vice versa.

**Table 1 jcm-11-00736-t001:** Median and interquartile range (IQR) of axiological characteristics of the 7 GH-treated patients.

	Baseline	Follow-Up	Relative Variation	*p*-Value
Weight (kg)	27.10 (22.40–33.10)	28.20 (23.70–32.30)	0.41 (0.16–0.46)	0.043
Height (cm)	124.60 (118.00–137.80)	127.50 (121.30–141.50)	0.23 (0.18–0.32)	0.018
SDS Height	−2.27 (−2.73–−1.76)	−2.19 (−2.25–−1.68)	−0.46 (−0.18–0.18)	0.075
BMI (kg/m^2^)	16.40 (15.30–17.80)	16.10 (15.10–17.50)	−0.13 (−0.18–0.00)	0.246
IGF1 (ng/mL)	152.90 (121.50–167.00)	266.50 (189.40–371.00)	0.84 (0.28–1.51)	0.018

The mean age of enrolled children is 10.9 years.

**Table 2 jcm-11-00736-t002:** Mean and standard deviation (SD) of diameters of 456 analyzed nevi.

	Baseline	Follow-Up	Relative Variation	*p*-Value
**Diameter Maximum S**	**2.22 ± 1.07**	**2.35 ± 1.10**	**0.07 ± 0.12**	**<0.001**
Diameter Maximum C	2.30 ± 0.47	2.38 ± 0.81	0.02 ± 0.07	0.12
**Diameter Minimum S**	**1.72 ± 0.85**	**1.85 ± 0.88**	**0.09 ± 0.13**	**<0.001**
Diameter Minimum C	1.79 ± 0.66	1.86 ± 0.75	0.03 ± 0.08	0.27

S: study group, C: control group.

**Table 3 jcm-11-00736-t003:** Mean and standard deviation (SD) of 222 nevi analyzed.

	Baseline	Follow-Up	Absolute Variation	*p*-Value
Asymmetry	0.26 ± 0.59	0.34 ± 0.57	0.07 ± 0.64	0.091
Borders	0.03 ± 0.10	0.05 ± 0.14	0.02 ± 0.14	**0.032**
Colors	0.62 ± 0.22	0.71 ± 0.26	0.09 ± 0.26	**<0.001**
Structures	0.77 ± 0.28	0.86 ± 0.28	0.09 ± 0.30	**<0.001**
ABCD Score	1.69 ± 0.84	1.96 ± 0.86	0.27 ± 0.87	**<0.001**

**Table 4 jcm-11-00736-t004:** Correlation coefficients between variations in the auxological with respect to the morphological parameters.

	Diameter Max	Diameter Min	Asymmetry	Borders	Colors	Structures	ABCD Score
Weight (kg)				R = 0.188*p* = 0.005	R = 0.255*p* < 0.001		
Height (cm)	R = −0.212*p* = 0.002						
SDS Height		R = −0.146*p* = 0.030				R = 0.162*p* = 0.015	R = 0.151*p* = 0.025
BMI (kg/m^2^)	R = 0.266*p* < 0.001	R = 0.142*p* = 0.034		R = 0.200*p* = 0.003	R = 0.346*p* < 0.001	R = 0.157*p* = 0.019	R = 0.268*p* < 0.001
IFG1 (ng/mL)	R = −0.267*p* < 0.001	R = −0.242*p* < 0.001

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
