# Peer review of "Effects of Growth Hormone (GH) Supplementation on Dermatoscopic Evolution of Pigmentary Lesions in Children with Growth Hormone Deficiency (GHD)"

_jcm, 2022, doi:10.3390/jcm11030736_

Round 1

Reviewer 1 Report

Effects of growth hormone supplementation on dermatoscopic evolution of pigmentary lesions….., by Panarese et al.

This is a timely and very nice study. I have listed below suggestions for grammatical and presentation improvements and some comments about the interpretation of the data.

INTRODUCTION

Line 1: “that switched”, change to “that have switched”.

The first paragraph needs to be rewritten, the second and third sentences are quite repetitive. The most important thing to emphasize is that melanoma is a health issue because of its propensity to metastasize, melanomas are heterogeneous in various ways as written in the paragraph, and importantly that a strong risk factor (the strongest?) for the development of melanoma is the presence of nevi, particularly large atypical growing nevi. 

Line 10: “increasing trend in men”. Better to just say “increasing incidence in men?”or rewrite the sentence in some other way. Then; “the difference is to be referred”….maybe it should something like..”this difference may be due to physiological differences in…..”, or “is postulated to be due to?”….

Introduction, page 2. Line 2: “had been” should be “have been”.

Page 2 line 8. “Melanoma cell cycle”. This is a little confusing as written as melanoma cell and cell cycle are phrases and one can’t write “melanoma cell cell cycle”!  Maybe the authors could link it to the previous part of the sentence, e.g. “human melanoma biopsies, in which the cell cycle was also considered to be under control…..”.

MATERIALS and METHODS

Line 1: change “belonging to” to “attending”.

Line 2: Define “SS”.

Line 9: “phototype” should be “skin phototype”?

In the middle of the paragraph: there is no need to describe SD.

Last line of the methods: “searched” should be “analyzed”.

RESULTS

Line 1: enrolled children, change ”is” to “was”.

Table 1: It might be better to name the second column as “follow-up” or “patient follow-up” since the word control in the mind of the reader applies to the control/non-patient group, if I understand correctly. It would seem interesting to include this data also for the control group maybe as a supplementary file.

Line 15-16: “both minor and minor axis”. One should be “maximum” or similar??

Line 16: To make this easier for the reader the authors should clearly say that nevi did not increase in diameter in the control group, but they did in the patient group.

Table 2 and 3: Remove control from the heading of column 2.

Table 3. Did the ABCD values change in the control group over time?

Figure 1: Add (mm) to the y-axis. Better to remove the word “control” from the x-axis labels?

Figures 3 and 4: Label with T0 and T1. Presumably the latter is the top one (Also for Figure 4). Expand the legend a bit, at least to say they are dermoscopic images. Is it customary to include scale bars on dermoscopic images?? Particularly for Figure 4, the authors should describe the dermatological changes over time and what they mean.

DISCUSSION

Page 8, Line 4: “was” change to “has been”.

Page 8, Line 8: “However” or even “But” would be better than “On the contrary” in this sentence.

Line 9: Insert (mm) after these diameter values.

Page 9. Line 2: change “increases of” to “increased 0.27 units” or a similar description.

Page 9, Line 3: Change “significative” to “significant”. Also, should the authors state that “the changes are not clinically significant over the times frame examined”??

Page 9, line 8: Change “is has been demonstrated” to “we demonstrated”….

In this paragraph (page 9 paragraph 2) it is stated that nevus growth is synchronized with melanocyte maturation and body growth features. How does this synchronization (or lack of) proceed in the control group? Analysis of this data may strengthen the authors’ findings.

Page 9, paragraph 2, line 2: change to “we demonstrated” …..

Table 4: With respect to the various positive and negative correlations between auxological and dermoscopic features. It makes one wonder why ABCD score for instance is not correlated with height and weight but is with BMI. If it would help the manuscript, maybe a way to present this (a complex array of interactions) better would be to also present table 3 in terms of a heat map based upon the correlation values, and including the non-significant correlations, for instance as an example red for degree of positive correlation and blue negative? Would it be possible/useful to construct such a heatmap for the control group also? Table 3 could be left as is and a heatmap figure added?

Page 9 Paragraph 3: The inverse correlation with IGF1 is very interesting and the paragraph should be expanded further for readers not very familiar with the HGH-IGF1 mechanisms. Maybe the authors could make more of this finding?

Page 9 paragraph 4: suggestion for change to first sentence to - “….In our study we recorded only minimum changes over time in the melanocytic lesions”…… The authors then state that figures 3 and 4 show examples but these dermoscopic changes are not pointed out in the figure(s) or discussed in the Results section.

Discussion last sentence: “normal physiological process”: do the authors mean “benign physiological process”??

CONCLUSION

This could be written a little more clearly to emphasize that GH therapy results in only relatively benign changes to nevi, and hence the safety profile with respect to nevus progression seems good. “detected at dermatoscopic but not the clinical level”, could this be clarified?, The changes cannot be detected by eye?

Author Response

Thank you very much for your  review .

here the changes according to the reviewers comment :

1) we have changed sentence in lane 1 

2) we have amended the paragraph  as requested and underlined the importance of metastatic potential 

3) line 10 : the sentence has been rephrased accordingly to suggestion : " increased incidence in men "

4) Page 2 line 8:  we have rewritten  completely the sentence  we hope that now is more understandable 

MATERIALS and METHODS

5) lane 1 , 2 and 9 have been change accordingly 

6) we have deleted the definition of sd 

7) analyzed  added

RESULTS

8) lane 1 changed accordingly

9) table 1 : we  corrected : follow up  we did not provide the same information since the control group as we stated in the M&M section was carefully matched for all baseline variables.

10)Line 15-16 we amended the wording ( major axis)

11)  lane 16 we have  made simpler the sentence   and rephrased  accorduingly to suggestion :" however their increase  was at follow up was  negligible  and not significant"

12) Table 2 and 3 corrected  the heading 

13) Table 3:  we explained in the text that "Since no significant differences were found in the control group, qualitative analyzes were performed only for the study group"   

14) Fig. 1  amended as requested  ( we amended also fig 2   with follow up instead of control)

15) fig. 3 and 4  we made the suggested changes  describing the demoscopica features  and labeling the figures 

Discussion 

16) page 8 lane 4 8 and 9  and page 9 lane 2 : we changed the sences accordingly and we added "Points" as the measuring unit 

17) page 9 paragraph 2 : we have added a sentence to clarify our hypothesis that GH  supplementation could have had a "pending" effect both on body growth and  melanocyte  maturation  moreover  we used the term "synchronization"  to emphasize how both  the values ​​of body growth and mole maturation in patients with GH deficit, are rapidly modificated by treatment, reaching the physiological values ​​of healthy patients (control). Since the evaluation of a pathological nevus is also based on its rapid evolution our work wants to determine the benignity of this process induced by GH suplementation.

18) table 4 

Thanks for the valuable suggestion, as regards the possibility of creating a heat map based upon the correlation values ​​also for control group we do not have all the necessary data available. In fact as previously stated, the control group immediately showed no statisically significant difference in the dimensional data of the moles,so  we focused on the study group. we understand that it would be more immediate and correct as a method of presentation but we do not have the possibility to present it correctly and completely.

as regards the interesting suggestion of BMI  it is necessary to evaluate how this index brings together in a mathematical relationship the two data(weight, height) which saparately are not significant but together may become.

19) page 9 paragraph 3 : we added a sentence on the role of the HGH-IFG1 axis in melanoma 

20 ) page 9 paragraph 4: corrected accordingly  and added demoscopica description in figure headings

21) eliminated the wording normal  that made the sentence confusing 

Conclusions

22)  we changed the phrase   clarifying accordingly to revier suggestions :  

(i.e. macroscopically by eye examination) and thus be considered only as begnign changes. The main limitation of this study is related to the reduced number of patients and the short 12-month follow-up.

Reviewer 2 Report

Authors presents interesting changes nevi observed in children Treated with GH therapy. They demonstrate significant changes in the intervention group.

I have a few observations. 

Authors state that GH present a good safety profile.  This statement should be modified as sample was low, follow up is short and there were   significant changes. Safety of effects of GH on melanocytes nevi cannot be assessed with this protocol.

On the same line, the last paragraph and conclusion should be based on the results and not the interpretation of the authors. Eg what do they mean with mole activation? Do they mean melanocytic activation?

There are important significant changes that are not mentioned. 

Minor :

Should clarify in figures which image correspond to pre and post treatment , images should be labeled, recommend to highlight changes and mention diameters. 

Author Response

1)  we added a  sentence in the conclusions  acknowledging the limitation of the study  that the sample is indeed  small   and the follow up limited to one year. as such : "The main limitation of this study is related to the reduced number of patients and the short 12-month follow-up so it would be necessary a longer follow up and an higher number of patients to definitively asses GH safety  in regard to melanoma formation ".

However we feel that the conclusion on melanocytic activation as evidenced by ABCD changes may prompt the cautious optimism on the safety of GH at least in our study period .

2) we corrected the headings in figures and added measurement units 

Reviewer 3 Report

Congratulations for your work. The interesting topic of the paper falls within the scope of MDPI. You have highlighted the aims, significance and the novelty of your work.

- Please correct the names of the authors according to MDPI Submission Guidelines: F. Panarese as Fabrizio Panarese

- Please fill the "Citation:" according to MDPI Submission Guidelines: Lastname, F.; Lastname, F.;

- Please add Polidori, N.; in "Citation:"

- Please reformulate the following sentences:

1. Other genetic and phenotype factors were thought to be in-volved in melanoma development, like: gender, age, skin type, number of nevi (>50 moles—high risk), family history, immune status, etc. [7,8]

found in: 

D Coricovac · 2018 · Cited by 49 — Other genetic and phenotype factors were discussed to be involved in melanoma development, like: gender, age, skin type, number of nevi (>50 moles—high ...

https://www.mdpi.com/1422-0067/19/6/1566/htm

2. The presence of growth hormone receptor (GHR) RNA in human skin cells, especially melanocytes, was reported more than 20 years ago [17], followed by the identification of autocrine levels

found in:

R Basu · 2017 · Cited by 33 — Moreover, primary human melanoma specimens were even found to have high levels of GH releasing hormone (GHRH) receptor (GHRHR) [21], while GHRH-analogs were ...

https://www.ncbi.nlm.nih.gov/pmc/articles/PMC5400608/

3. Moreo-ver, primary human melanoma specimens were found to have high levels of GH releasing hormone (GHRH) receptor (GHRHR) [25], while GHRH-analogs were successful in sup-pressing malignant melanoma growth in vivo [26].

found in:

Potential Role of Human Growth Hormone in Melanoma ...

https://jamanetwork.com/journals/jamadermatology/fullarticle/1379282

Author Response

We have amended all the typos and MDPI submission guidelines derangement 

We have changed the sentences that presented similarities with other in literature.

Round 2

Reviewer 2 Report

Cannot conclude safety of GH therapy on nevi based on this study.

Author Response

We would like  again to thank the reviewer for the rapid response .

Since there are (to our knowledge)  no evidences in literature  of melanoma development in children  treated with GH and our study  follow up was not so brief ( 12 months) we felt  that it was sound to write  that GH could be a safe treatment in this regard. We added this sentence , after the first review  : 

"The main limitation of this study is related to the reduced number of patients and the short 12-month follow-up so it would be necessary a longer follow up and an higher number of patients to definitively asses GH safety  in regard to melanoma formation ."  

In the light of this new comment  we  modified the first part of the conclusions as such : 

From our data emerges that GH therapy  does not  induce  drastical changes in moles, and that the effects of such therapy are probably exclusively related to a melanocyte activation that can be detected at dermatoscopic but not at clinical level (i.e. macroscopically by eye examination) within one year from the beginning of the treatment . The main limitation of this study is related to the reduced number of patients and the short 12-month follow-up so it would be necessary a longer follow up and an higher number of patients to definitively asses GH safety  in regard to melanoma formation.   we have thus eliminated the  sentence  supporting the safety of gh we hope that this change  would allow the paper to be considered for publication